# A clinical SARS-CoV-2 M^pro inhibitor blocks replication of multiple enteroviruses and confers oral in vivo protection in animal models

Zhengyu Ye[1,2,☺], Wenhao Dai[3,4,☺], Shaolin Zhang[5,6,☺], Yingchun Xiang[5,6,☺], Jinlin Wang[3], Yumin Zhang[1,2], Wenyuan Cao[5,6], Johan Neyts[7], Zuyi Li[8], Fan Feng[1,9], Gengfu Xiao[1,2], Hong Liu[3,4*], Junyuan Cao[1,2*], Lei-ke Zhang[1,2,6*]

**1** State Key Laboratory of Virology, Wuhan Institute of Virology, Center for Biosafety Mega-Science, Chinese Academy of Sciences, Wuhan, China, **2** University of the Chinese Academy of Sciences, Beijing China, **3** State Key Laboratory of Drug Research, Shanghai Institute of Materia Medica, Chinese Academy of Sciences, Shanghai, China, **4** School of Pharmaceutical Science and Technology, Hangzhou Institute for Advanced Study, University of Chinese Academy of Sciences, Hangzhou, China, **5** HuaWu Biotech, Wuhan, China, **6** Hubei Jiangxia Laboratory, Wuhan, China, **7** University of Leuven, Leuven, Belgium, **8** Huazhong University of Science and Technology, Wuhan China, **9** School of Chemical Engineering and Pharmacy, and Hubei Key Laboratory of Novel Reactor and Green Chemical Technology, Wuhan Institute of Technology, Wuhan China

☺ These authors contributed equally to this work
* hliu@simm.ac.cn (HL); caojunyuan@wh.iov.cn (JC); zhangleike@wh.iov.cn (LZ)

## Abstract

*Enteroviruses,* which belong to the family *Picornaviridae*, cause hand, foot, and mouth disease (HFMD), respiratory symptoms, and severe neurological complications in children. Since vaccines cannot provide cross-protection against different serotypes of enteroviruses, the development of broad-spectrum anti-enteroviral drugs is imperative. The viral 3C protease (3C^pro), which is essential for polyprotein processing represents a validated target for therapeutic intervention. Importantly, enterovirus 3C^pro shares conserved structural and catalytic features with coronavirus main protease (M^pro, also known as 3C-like protease, 3CL^pro), providing a rationale for cross-target inhibitor repurposing. Through targeted screening of peptidomimetic protease inhibitors, a clinical-stage SARS-CoV-2 M^pro inhibitor was identified as a potent inhibitor of enterovirus A71 (EV71) 3C^pro. Bofutrelvir displayed nanomolar antiviral activity in multiple cell lines and demonstrated broad-spectrum efficacy against several enteroviruses including coxsackievirus B5, coxsackievirus A16 (CA16) and echovirus 11. In EV71 infected neonatal mice, intraperitoneal administration of bofutrelvir markedly reduced viral loads in brain, spinal cord, and muscle, alleviated clinical symptoms, and suppressed tissue inflammation. Oral administration of bofutrelvir also provided therapeutic benefits in neonatal mice models of both EV71 and CA16. Crystallographic analysis revealed that bofutrelvir binds in the conserved

**Data availability statement:** All relevant data are within the paper and its Supporting Information files.

**Funding:** This work was supported by grants from National Key Research and Development Plan of China (2024YFC2607300 to G.X), National Natural Science Foundation of China (82130105 to H.L, Y. Z, 82341091 to W.D and U22A20379 to G.X), National Key Research and Development Plan of China (2022YFC2303300 to L.Z), the Key R&D Program of Hubei Province (2021BCD004 to L.Z). The funders had no role in study design,data collection and analysis, decision to publish, or preparation of the manuscript.

**Competing interests:** The authors have declared that no competing interests exist.

substrate-binding cleft of EV71 3C$^{pro}$, elucidating its molecular mechanism of inhibition. These findings identify bofutrelvir as a broad-spectrum peptidomimetic 3C$^{pro}$ inhibitor with strong antiviral efficacy against enteroviruses and highlight its potential for repurposing as a promising antiviral candidate for the treatment of enteroviral infections.

## Author summary

Hand, foot and mouth disease (HFMD) caused by enteroviruses remains a major public health concern, yet no approved antiviral therapy is currently available. Although SARS-CoV-2 M$^{pro}$ and EV71 3C$^{pro}$ share low sequence similarity, they are highly conserved in overall three-dimensional fold, catalytic cysteine-based mechanism, and strict specificity for P1-glutamine substrates. This structural and functional homology provides a rationale for repurposing coronavirus protease inhibitors to target enteroviral proteases. In this study, we identify bofutrelvir, a clinical-stage protease inhibitor originally developed for SARS-CoV-2, as a potent and broad-spectrum inhibitor of enterovirus. We show that bofutrelvir effectively blocks the activity of the enteroviral 3C$^{pro}$, a key enzyme required for viral replication. Using structural, cellular, and animal models, we demonstrate that bofutrelvir suppresses viral replication, reduces disease severity, and provides protection against EV71 and CA16 infections in neonatal mice following oral administration. Importantly, bofutrelvir exhibits favorable pharmacokinetic properties and has previously demonstrated good safety and tolerability in human clinical trials. Together, our findings highlight bofutrelvir as a promising candidate for repurposing as an orally available antiviral therapy for enterovirus-associated diseases.

## Introduction

Enteroviruses (EVs) belong to the genus *Enterovirus* within the *Picornaviridae* family, non-enveloped, positive-sense single-stranded RNA viruses [1,2]. The genus *Enterovirus* comprises major human pathogens, such as poliovirus (PV), coxsackievirus (CV), echovirus, emerging numbered enteroviruses like enterovirus-A71 (EV71) and enterovirus-D68 (EV68), and rhinoviruses (RVs). Hand, foot, and mouth disease (HFMD) is a common infectious disease in children under 5 years old, caused by enteroviruses. EV71 and Coxsackievirus A16 (CA16) are the most prevalent pathogens, with EV71 responsible for the majority of severe and fatal cases, as it is associated with serious neurological complications such as encephalitis and acute flaccid paralysis [3,4]. EV71 can invade the central nervous system (CNS) via the bloodstream or through retrograde axonal transport along peripheral nerves, leading to infections in motor neurons and other neural pathways [5–8]. Although

most EVs infections are self-limiting, their high prevalence and potential for severe outcomes underscore the urgent need for effective antiviral therapies [1,9].

The EVs viral life cycle presents multiple targets for intervention, including attachment, uncoating, translation, polyprotein processing, assembly, and release. Following entry, the viral RNA is translated into a single polyprotein, subsequently cleaved by viral proteases 2A$^{pro}$ and 3C$^{pro}$ into structural (P1) and non-structural (P2 and P3) proteins. The P1 region yields capsid proteins VP1–VP4, while P2 and P3 produce proteins essential for replication, including 2A$^{pro}$, 2B, 2C, 3A, 3B (VPg), 3C$^{pro}$, and 3D$^{pol}$ [10]. Several antiviral agents targeting different stages of the EVs life cycle have been investigated [1,11–13]. Inhibiting early steps has been explored, with monoclonal antibodies targeting EVs showing promise in preclinical studies [14,15]. A monoclonal antibody against EV-D68 has entered Phase I clinical trials (NCT06444048). Pleconaril and Vapendavir are capsid-binding compounds that bind a hydrophobic pocket in VP1, preventing viral uncoating and RNA release [16,17]. However, their clinical development has been hindered by issues such as viral resistance and limited efficacy (NCT00031512, NCT04838145, NCT06149494). Nucleoside analogs like 4'-fluorouridine (4-Flu) [18] and NITD008 [19,20] target the viral RNA-dependent RNA polymerase (3D$^{pol}$), demonstrating potent antiviral activity in vitro and in animal models.

The 3C$^{pro}$, a chymotrypsin-like cysteine protease, plays a pivotal role in polyprotein processing and modulating host cell functions, making it an attractive target for antiviral development [10,21–24]. Enterovirus 3C$^{pro}$ and coronaviruses M$^{pro}$ exhibit similarities in overall fold and catalytic architecture (including conserved cysteine-based active site and substrate-recognition architecture) [25,26]. Rupintrivir (AG7088) is a first-generation 3C$^{pro}$ inhibitor initially developed for human rhinovirus infections [24,27,28]. It features a γ-lactam moiety that mimics the glutamine residue at the P1 position of the viral 3C$^{pro}$ substrate, enabling effective inhibition of RVs replication [29]. Subsequently, rupintrivir was repurposed for EV71 inhibition and demonstrated in vivo antiviral efficacy [30–32]. Based on its structural series 3C$^{pro}$ inhibitors were developed, including the NK-1.8k series [33,34], SG85 series [16,35], and compound 18p, 28f [36,37], each exhibiting improved potency.

Peptidomimetics, synthetic compounds that mimic the structure and function of peptides, have been successfully employed in antiviral drug development [38]. Such compounds often offer improved stability and bioavailability compared to natural peptides. Notable examples include HIV protease inhibitors (darunavir, lopinavir/ritonavir) [39,40] and HCV NS3/4A protease inhibitors [41–44]. The strategy has also been applied to coronaviruses, leading to the development of SARS-CoV-2 M$^{pro}$ inhibitors like nirmatrelvir and simnotrelvir [45–47].

In this study, we identified bofutrelvir (FB2001) as a potent inhibitor of EV71 3C$^{pro}$ through enzymatic and cellular screening. Bofutrelvir, a peptidomimetic aldehyde inhibitor, was identified at an early stage during the SARS-CoV-2 pandemic and advanced into Phase II/III clinical trials (NCT05675072, NCT05445934) [48,49], where it demonstrated favorable safety and tolerability. Our findings reveal that bofutrelvir can be effectively repurposed as a promising anti-enterovirus candidate. While sharing a similar scaffold with 18p, bofutrelvir is structurally distinct at the P2 position, where a cyclohexyl group replaces the planar aromatic moiety found in 18p.

Notably, bofutrelvir exhibited broad-spectrum antiviral activity against multiple human enteroviruses, with half-maximal effective concentration (EC$_{50}$) values in the nanomolar range. In a neonatal mouse model, intraperitoneal administration of bofutrelvir provided significant protection against enterovirus infection. Furthermore, pharmacokinetic data in dogs supported its oral bioavailability, and we demonstrated that oral administration of bofutrelvir effectively ameliorated clinical symptoms in infected mice. Mechanistically, we resolved the co-crystal structure of bofutrelvir in complex with EV71 3C$^{pro}$, providing structural insight into its antiviral mode of action. Structural analysis indicates that this flexible P2 moiety adopts a distinct binding mode within the S2 pocket, potentially enabling improved target engagement. These findings suggest that bofutrelvir has the potential to be repurposed as a broad-spectrum antiviral candidate against human enteroviruses.

## Results

### High-throughput screening identifies potent inhibitors of EV71 3C^pro

Peptidomimetic protease inhibitors represent a key structural class in the rational design of antiviral agents targeting viral proteases. To explore the potential of these inhibitors against EV71, we constructed a focused library enriched in peptido-mimetic compounds. We searched the PubChem database using the keyword "protease inhibitor" and manually curated a subset of compounds featuring canonical peptidomimetic motifs. A total of 104 structurally diverse peptidomimetic protease inhibitors were selected and classified into major categories based on their known target protease classes, including cysteine, serine, and aspartic as shown in Fig 1A. All selected compounds were dissolved in DMSO at 40 mM stock concentration and stored at –80 °C for screening. The compounds were listed in S1 Table. To evaluate the inhibitors' potential, a primary enzymatic screen was performed using a fluorescence resonance energy transfer (FRET)–based assay on the recombinant EV71 3C^pro. The screen was performed at a concentration of 100 μM for all 104 inhibitors (Fig 1B). Rupintrivir, a known 3C protease inhibitor originally developed for rhinovirus, was used as a positive control [30,31].

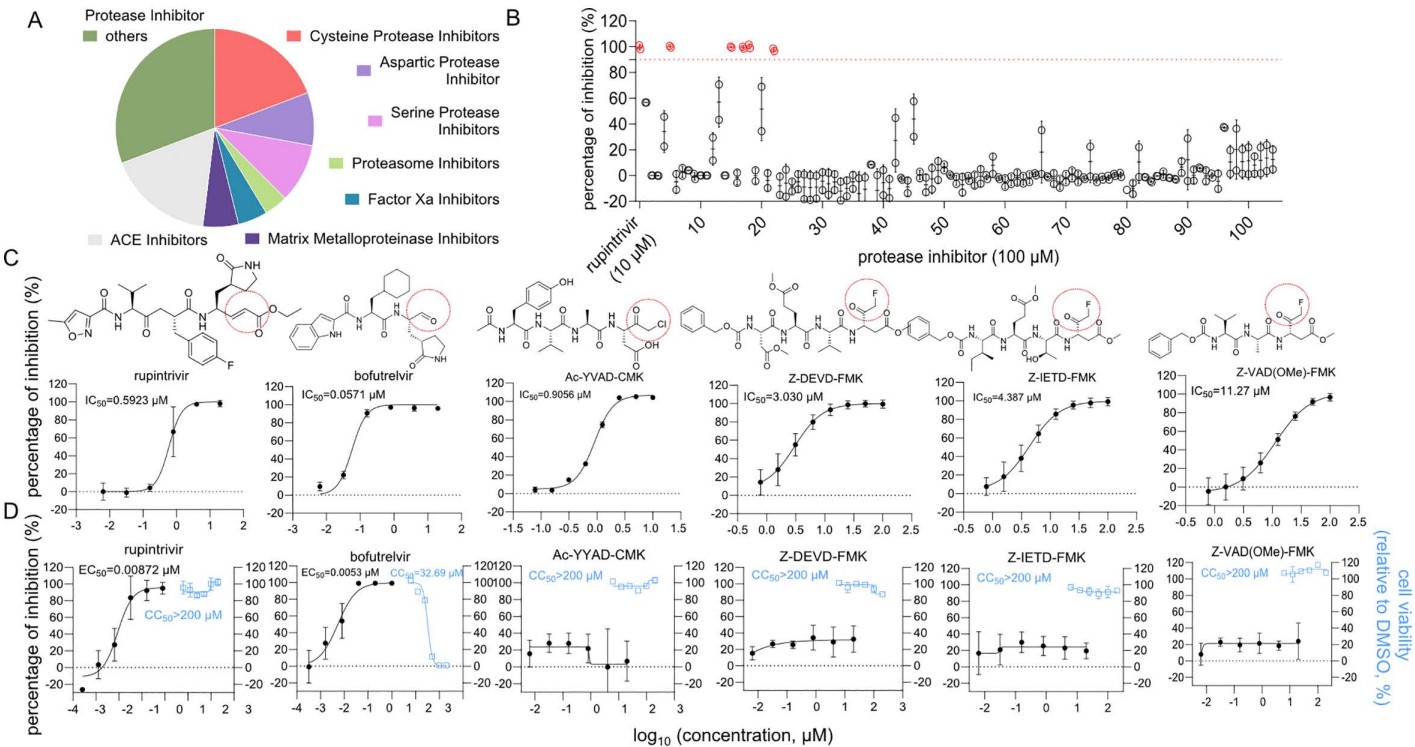

**Fig 1. High-throughput screening identifies potent inhibitors of EV71 3C^pro. A.** Composition of the protease inhibitor library comprising 104 compounds, categorized into cysteine protease inhibitors (20), aspartic protease inhibitors (9), serine protease inhibitors (10), proteasome inhibitors (4), Factor Xa inhibitors (5), ACE (angiotensin-converting enzyme) inhibitors (17), and others. **B.** Primary screening of the compound library based on EV71 3C^pro enzymatic activity. Exclude rupintrivir five compounds exhibiting >90% inhibition at 100 μM were identified as primary hits. **C.** Chemical structures and dose-response curves of selected inhibitors with significant EV71 3C^pro inhibitory activity. The chemical structures are shown above each dose-response curve. Red circles highlight the electrophilic warheads (e.g., aldehyde, fluoromethyl ketone) which are predicted to form covalent interactions with the catalytic cysteine residue of the 3C^pro. Inhibition was quantified using fluorescence-based enzymatic assays, and IC_50 values were calculated accordingly. **D.** Antiviral activity and cytotoxicity of selected hits in RD cells. Black curves represent antiviral efficacy by using RT-qPCR assay (reduction of viral replication), while blue curves indicate cytotoxicity by using CCK-8. Compounds bofutrelvir demonstrated potent antiviral activity (EC_50 = 0.0053 μM) with cytotoxicity (CC_50 = 32.69 μM). Data represent mean±SD from three independent experiments.

Among the tested compounds, five protease inhibitors showed dose-dependent activity against EV71 3C$^{pro}$, indicating potential for further evaluation as antiviral candidates. Among these active hits, bofutrelvir stood out as a leading candidate. Originally developed as an antiviral targeting coronaviral M$^{pro}$, bofutrelvir features a γ-lactam moiety at the P1 position, similar to rupintrivir. This γ-lactam effectively mimics the glutamine side chain recognized by both EV71 3C$^{pro}$ and SARS-CoV-2 M$^{pro}$, enabling strong binding to the S1 pocket. Unlike rupintrivir, which contains a Michael acceptor warhead, bofutrelvir uses an aldehyde group as its reactive electrophile. This aldehyde reacts with the protease catalytic cysteine, forming a reversible covalent adduct. This structural trait likely contributes to its high potency across various cysteine proteases. Besides bofutrelvir, several caspase-targeting inhibitors also showed significant enzymatic inhibition. Notably, Ac-YVAD-CMK, Z-DEVD-FMK, Z-IETD-FMK, and Z-VAD(OMe)-FMK—originally developed as caspase inhibitors—demonstrated dose-dependent inhibition of EV71 3C$^{pro}$ activity *in vitro*. These compounds share electrophilic warheads such as chloromethyl ketone (CMK) or fluoromethyl ketone (FMK), which covalently modify active-site cysteine residues.

To determine whether the enzymatic inhibition observed *in vitro* translated to antiviral efficacy in a cellular context, we next evaluated the five hit compounds in EV71-infected RD cells using Reverse transcription quantitative Real-time PCR (RT-qPCR) antiviral assay. The results are shown in Fig 1D, where both antiviral activity (black curves) and cytotoxicity (blue curves) were quantified across a range of compound concentrations. Among all tested candidates, only bofutrelvir exhibited robust antiviral activity in cells, with an EC$_{50}$ of 0.0053 µM and minimal cytotoxicity.

In contrast, the other four caspase inhibitors, although effective at inhibiting EV71 3C$^{pro}$ enzymatic activity *in vitro*, failed to demonstrate measurable antiviral effects in cell-based assays. One possible explanation for this discrepancy is that these compounds preferentially bind to endogenous caspase family proteases within the host cells, which share similar catalytic cysteine residues. Such non-selective engagement may reduce their intracellular availability for targeting the viral 3C$^{pro}$, thereby diminishing their antiviral efficacy. In addition to potential preferential binding to endogenous caspases, limited cellular permeability may also contribute to this discrepancy, as the negatively charged aspartic acid residues could hinder membrane penetration and restrict intracellular access to viral 3C$^{pro}$.

## Bofutrelvir exhibits potent cellular antiviral activity and broad-spectrum efficacy against multiple *Enteroviruses*

To evaluate the cellular antiviral activity of bofutrelvir, we tested its efficacy against EV71 in three distinct cell lines: human glioma U251 cells, African green monkey kidney Vero cells, and human intestinal epithelial Caco-2 cells. These cell types were chosen to reflect EV71's tissue tropism: the virus can invade the CNS (causing encephalitis) and initiate infection in the gastrointestinal tract. Across all three cell types, bofutrelvir demonstrated consistent, dose-dependent antiviral activity, with half-maximal cytotoxic concentrations (CC$_{50}$) exceeding 100 µM, indicating a favorable safety profile (Fig 2A). To further evaluate the compound's antiviral effectiveness at preventing infectious virus production, we measured viral titers in the supernatant of infected Vero cells using the TCID$_{50}$ assay. Notably, treatment with 0.04 µM bofutrelvir caused approximately a 1-log reduction in viral titer, from $6 \times 10^5$ to $6 \times 10^4$ TCID$_{50}$/ml (Fig 2B). This level of reduction was similar to the suppression seen in RT-qPCR assays performed in RD cells. To visualize the effect on viral protein expression, we conducted immunofluorescence assays targeting the EV71 capsid protein VP1. As shown in Fig 2C, treatment with 0.032 µM bofutrelvir resulted in a significant decrease in VP1 signal intensity, with over 50% reduction compared to the virus-infected control. To clarify the stage at which the compound acts, we performed a time-of-addition assay in RD cells. Bofutrelvir maintained antiviral activity when administered after entry, suggesting it primarily disrupts the viral replication phase (S1 Fig).

Given the conserved nature of picornaviral 3C$^{pro}$, we next assessed the broad-spectrum potential of bofutrelvir against a panel of representative human enteroviruses: coxsackievirus B5 (CVB5); coxsackievirus A6, A10, and A16 (CA6, CA10, CA16); and echovirus 11 (Echo11). RT-qPCR quantification of viral RNA revealed potent inhibitory activity, with EC$_{50}$ values in the nanomolar range (Fig 2D). These findings indicate that bofutrelvir possesses broad-spectrum antiviral activity across the human enteroviruses, likely through conserved targeting of 3C$^{pro}$.

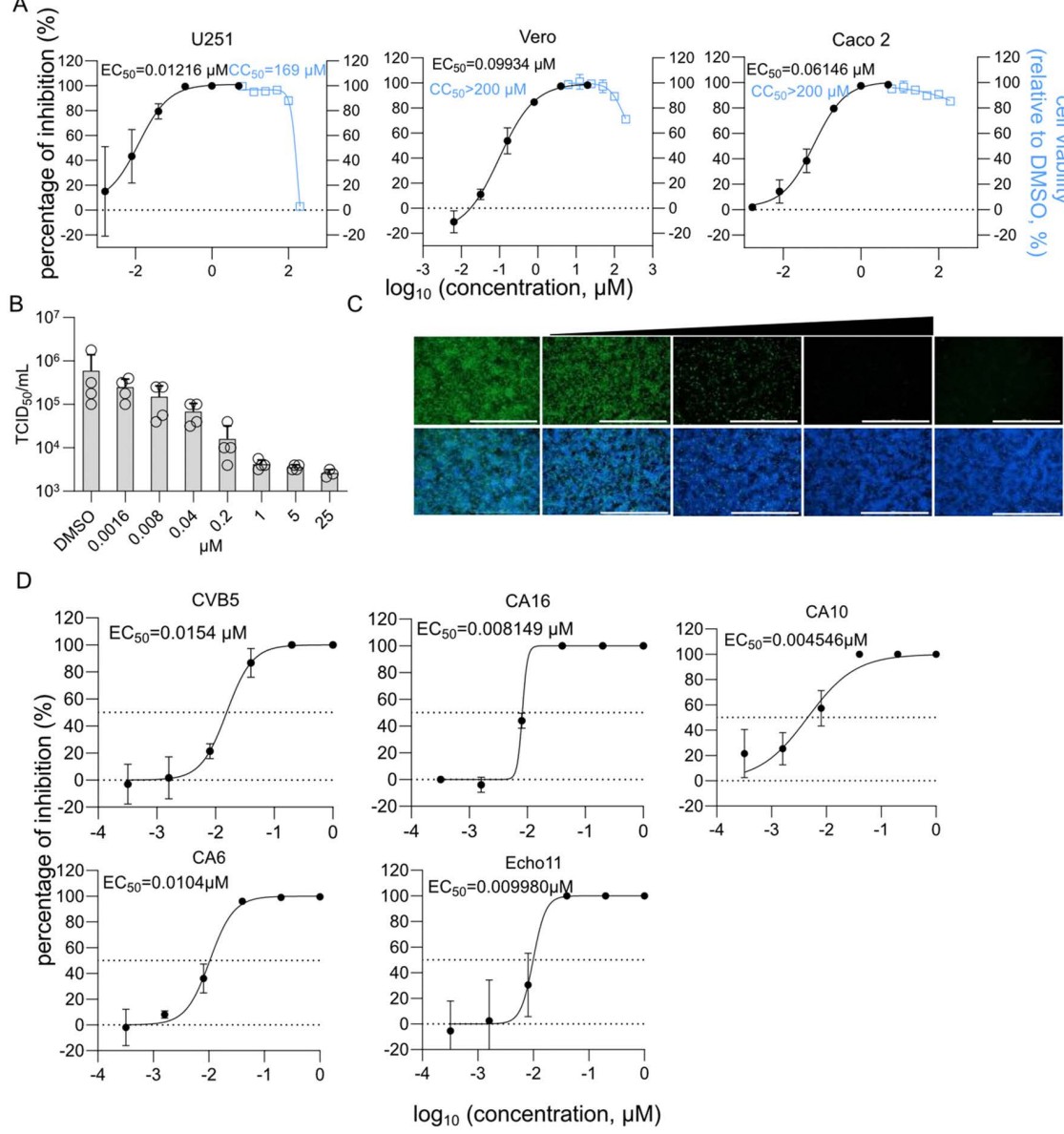

**Fig 2. Antiviral efficacy of bofutrelvir against EV71 and other enterovirus in vitro. A.** Bofutrelvir inhibited EV71 replication in a dose-dependent manner across three different cell lines (U251, Vero, and Caco-2), as determined by antiviral activity with different MOI (U251 MOI = 0.005, Vero MOI = 0.01, Caco-2 MOI = 1) and cytotoxicity assays. Black curves represent antiviral activity determined by RT–qPCR, while blue curves indicate cell viability. **B.** Dose-dependent effects of bofutrelvir inhibited EV71 replication in Vero cells as determined by viral $TCID_{50}$ assay. **C.** Immunofluorescence assay (IFA) to visualize the antiviral effect of bofutrelvir in EV71-infected RD cells. Viral capsid protein VP1 was detected using an anti-VP1 antibody (green), and nuclei were counterstained with DAPI (blue). Treatment with bofutrelvir resulted in a marked decrease in VP1 signal in a concentration dependent. **D.** Broad-spectrum antiviral activity of bofutrelvir against other enteroviruses, including CVB5, CA16, CA10, CA6, and Echo11. Dose-dependent effects of bofutrelvir on virus (MOI = 0.005-0.05) replication in RD cells as determined by RT–qPCR.

## Interaction between EV71 3C^pro and bofutrelvir

To elucidate the molecular mechanism underlying the antiviral activity of bofutrelvir, we determined the co-crystal structure of EV71 3C^pro in complex with bofutrelvir at high resolution 2.2 Å (PDB: 9VKK). The structure reveals that bofutrelvir binds

directly within the substrate-binding cleft of EV71 3C^pro, adopting an extended conformation and occupying the peptide substrate recognition pockets S1, S2, and S4, along with the catalytic site S1'(Fig 3A and 3B). The aldehyde warhead of bofutrelvir is positioned to contact the protease's catalytic cysteine (Cys147), consistent with the covalent interaction model observed for other cysteine protease inhibitors [22,24].

The P1, P2, and P3 positions of bofutrelvir were assigned based on its interactions with key residues lining the substrate-binding pockets. As illustrated in Fig 3C, the P1 moiety forms hydrogen bonds with His161 and Thr142 in the S1 pocket, anchoring the compound through polar contacts. In the S2 pocket (Fig 3D), the P2 group interacts with His40, Lys130, and Glu71, primarily through hydrophobic interactions. The P3 group, residing in the solvent-exposed S4 pocket (Fig 3E), forms specific interactions with Gly163 and Gly164. These structural insights delineate a multi-point interaction mechanism that enhances the potency of bofutrelvir. Structural superposition of the EV71 3C^pro–bofutrelvir and EV71 3C^pro–rupintrivir complexes reveals a conserved catalytic binding mode with distinct subsite engagement. The P1 γ-lactam moieties of both inhibitors overlap closely in the S1 pocket. In contrast, bofutrelvir's cyclohexyl P2 group inserts deeper into the hydrophobic S2 pocket and adopts a non-planar conformation, whereas rupintrivir's aromatic P2 group remains

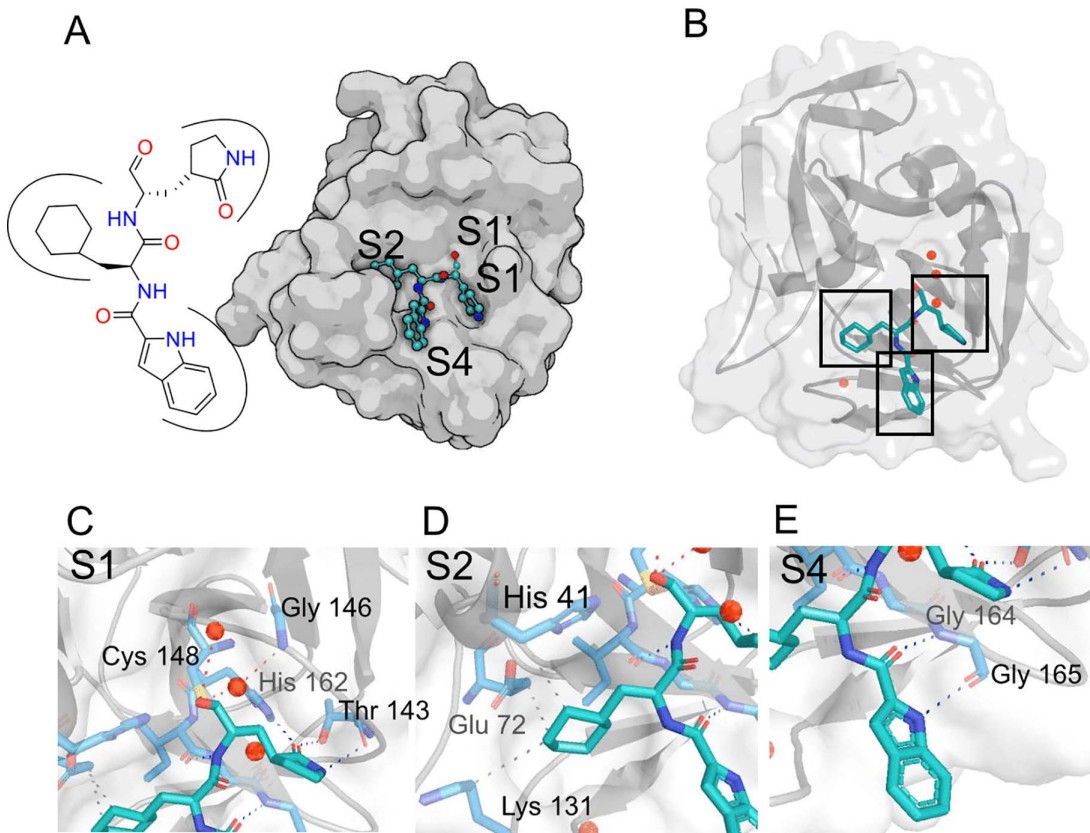

**Fig 3. Interaction between EV71 3C^pro and bofutrelvir. A.** Chemical structure of bofutrelvir (left) and its binding conformation within the active site of EV71 3C^pro (right). The three subsite-binding moieties (S1, S2, and S4) are indicated. **B.** Overall crystal structure of EV71 3C^pro in complex with bofutrelvir, shown as a ribbon model with partial surface representation. The black boxes highlight the S1, S2, and S3 subsite interactions between the inhibitor and the protease. **(C–E)** Close-up views of the S1 **(C)**, S2 **(D)**, and S4 **(E)** binding pockets, showing detailed interactions between bofutrelvir and key residues of EV71 3C^pro. In the S1 site, hydrogen bonds are formed with His161 and Thr142. The Gly 145 and Cys 147 chain interact with water; in the S2 site, interactions involve Glu71, His40 and Lys130; and in the S4 site, Gly163, Gly164 contribute to inhibitor binding. Hydrogen bonds are shown as yellow dashed lines. Water molecules are depicted as red solid circles.

closer to the pocket periphery. In addition, rupintrivir contains extended P3/P4 substituents, while bofutrelvir adopts a more compact binding mode (S2 Fig).

## Bofutrelvir suppresses EV71 replication and pathogenesis in a neonatal mouse model

To evaluate *in vivo* efficacy, we used a neonatal mouse model of EV71 infection [50]. Six-day-old ICR mice were infected intraperitoneally with $10^6$ PFU of EV71 and subsequently treated twice daily (bid) with bofutrelvir at 12.5, 25, or 50 mg/kg (mpk) body weight by intraperitoneal administration. Treatment began on the day of infection (Day 0) and continued for seven consecutive days. The nucleoside analog NITD008 (5 mpk, oral, bid) was included as a positive control. Mice were euthanized on Day 6, and tissues were collected for analysis (Fig 4A).

Clinical scores reflecting disease severity (e.g., limb paralysis and reduced mobility) were markedly improved in bofutrelvir-treated groups compared to vehicle controls (Fig 4B). Tissue samples from the brain, spinal cord, and hind limb muscle were harvested to quantify viral replication. As shown in Fig 4C–4E, RT-qPCR analysis revealed that bofutrelvir significantly reduced EV71 viral RNA levels in all tested tissues in a dose-dependent manner, with the 50 mpk group showing near-complete suppression. Consistently, viral titers measured by plaque assay (Fig 4F–4H) demonstrated strong inhibition of infectious virus production in both CNS and peripheral tissues, with bofutrelvir achieving viral clearance at 25–50 mpk doses comparable to the positive control. Notably, the 50 mpk group exhibited near-baseline scores throughout the observation period, indicating effective protection against EV71-induced neurological symptoms. Immunofluorescence staining of the viral capsid protein VP1 further demonstrated a dose-dependent reduction in viral protein expression in both brain and muscle tissues. VP1 was barely detectable in the 25 mpk group and was completely suppressed at 50 mpk, mirroring the effects of NITD008 (S3 Fig).

To assess the anti-inflammatory and neuroprotective effects of bofutrelvir *in vivo*, histopathological examinations and cytokine analyses were performed on EV71-infected neonatal mice. Hematoxylin and eosin (H&E) staining of brain sections revealed that vehicle-treated animals displayed characteristic signs of neuroinflammation, including vacuolar degeneration (Fig 5A, arrowheads). In contrast, bofutrelvir treatment alleviated these pathological features in a dose-dependent manner. mRNA levels of IL-1β, TNF-α, and IL-6 in brain tissue were significantly reduced by bofutrelvir in a dose-dependent manner, indicating suppression of neuroinflammation (Fig 5B–5D).

Consistent with the reduction in clinical symptoms and viral titer in the hind foot muscle at Fig 4H and 4E staining of hind limb muscle revealed that vehicle-treated mice exhibited extensive muscle fiber damage in the hind limb, whereas bofutrelvir treatment preserved muscle architecture in a dose-dependent manner (Fig 5E). Cytokine analysis further confirmed the local anti-inflammatory effect, as bofutrelvir markedly suppressed IL-1β, TNF-α, and IL-6 expression in muscle tissue (Fig 5F–5H). Together, these data highlight the ability of bofutrelvir to protect both neural and muscular tissues from EV71-induced pathology.

## Oral administration of bofutrelvir confers protective antiviral efficacy against EV71 and CA16 infection in neonatal mice

To evaluate the therapeutic potential of oral delivery, we first assessed the pharmacokinetic profile of bofutrelvir in beagle dogs following a single oral dose of 25 mpk. As shown in Fig 6A, bofutrelvir exhibited favorable oral bioavailability, with plasma concentrations sustained above the antiviral $EC_{50}$ and $EC_{90}$ values for an extended period.

An oral efficacy model in neonatal mice to evaluate the antiviral activity of bofutrelvir *in vivo* was established (Fig 6B include sample collection assay and survival curve). Six-day-old ICR mice were infected intraperitoneally with $10^6$ PFU of EV71 and treated via oral gavage with bofutrelvir at 25, 50, 75, or 100 mpk twice daily for seven consecutive days. NITD008 (5 mpk, oral) served as a positive control. Clinical scores were recorded daily to monitor disease progression (Fig 6I), and survival was followed until 14 days post infection. As shown in Fig 6I, Vehicle-treated mice developed severe neurological symptoms, whereas bofutrelvir-treated groups showed dose-dependent improvement. Mice receiving

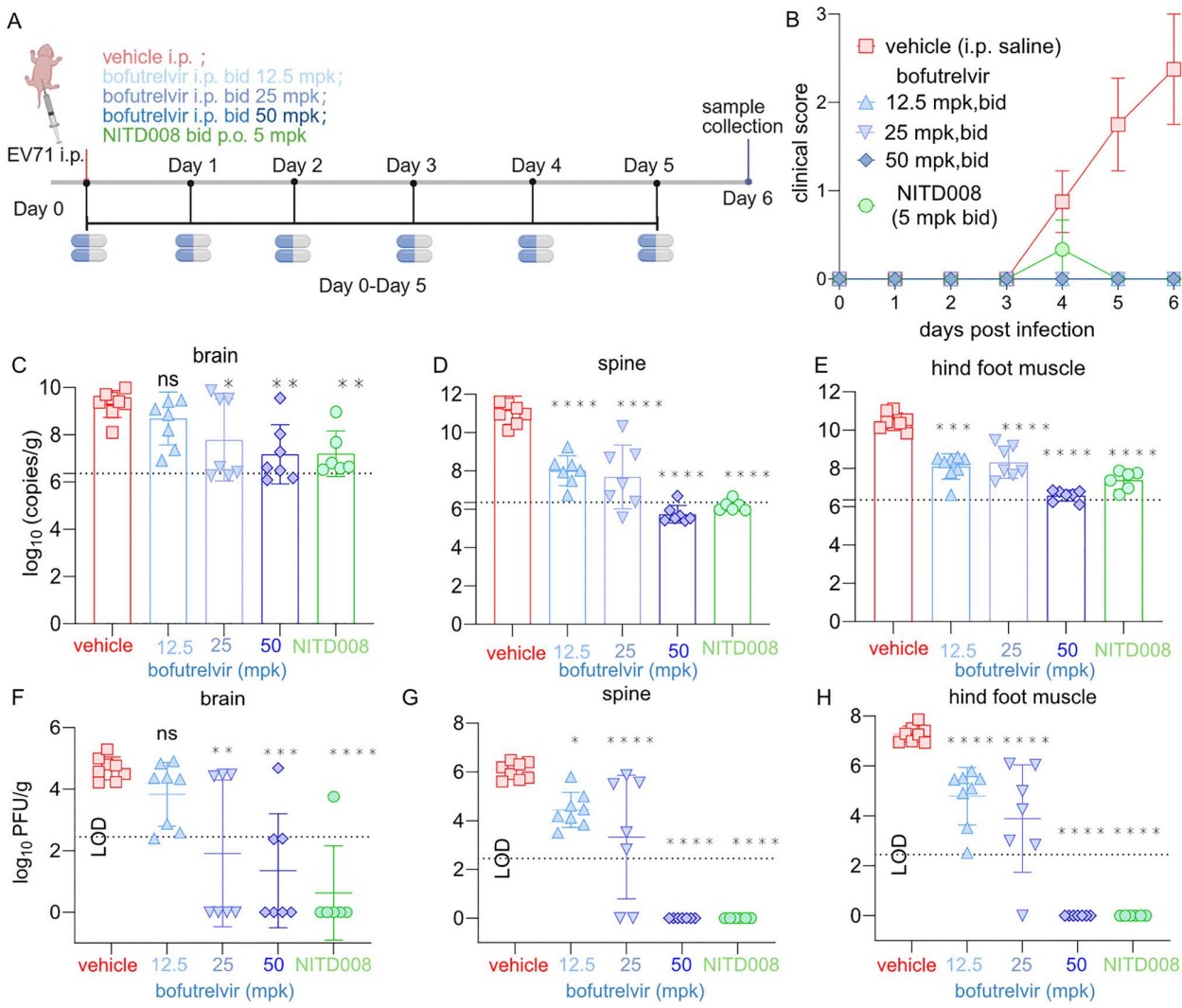

**Fig 4. Bofutrelvir suppresses EV71 replication and pathogenesis in a neonatal mouse model. A.** Schematic of the *in vivo* experimental design.6 days old neonatal ICR mice were infected with EV71 via intraperitoneal (i.p.) injection $10^6$ pfu on Day 0 and treated twice daily with bofutrelvir (12.5, 25, or 50 mpk, i.p.) or the positive control compound NITD008 (5 mpk, p.o.) from Day 1 to Day 6. Samples were collected on Day 6. **B.** Clinical scores of infected mice over time, showing treatment-dependent improvement in disease symptoms. (Clinical illness was scored as follows: 0, normalcy; 1, ruffled hair and hunchbacked appearance; 2, limb weakness; 3, paralysis in one limb; 4, paralysis in two limbs; 5, lose of the ability to move and ingest; 6, death. As mice with a clinical score of 5 usually die in 1 day, they were euthanized with carbon dioxide to reduce their suffering and their death days were accounted on the next day). Figure Created in BioRender. Cao, **J.** (2026) https://BioRender.com/yo37i97. C–E. Quantification of EV71 viral RNA copies numbers in brain, spinal cord, hind foot muscle, and hind foot tissues by RT-qPCR. F–H. Infectious viral titers measured in corresponding tissues by plaque assay. Data are shown as mean±SD from independent biological replicates (n=6-8). Scale bars, 1000 μm. Statistical significance was compared between the vehicle group and test groups, which represented by asterisks marked correspondingly in the figure. (* P < 0.05 and ** P<0.01, *** P<0.001 **** P<0.0001 by non-parametric One-way ANOVA analysis, respectively).

different concentrations of bofutrelvir displayed minimal clinical signs, with protection levels comparable to those seen in the NITD008 group.

To quantify antiviral effects, viral RNA copy numbers in the brain, spinal cord, and hind limb muscle were measured by qRT-PCR on Day 6 (Fig 6C-6E). Bofutrelvir treatment reduced viral RNA levels in all tissues in a

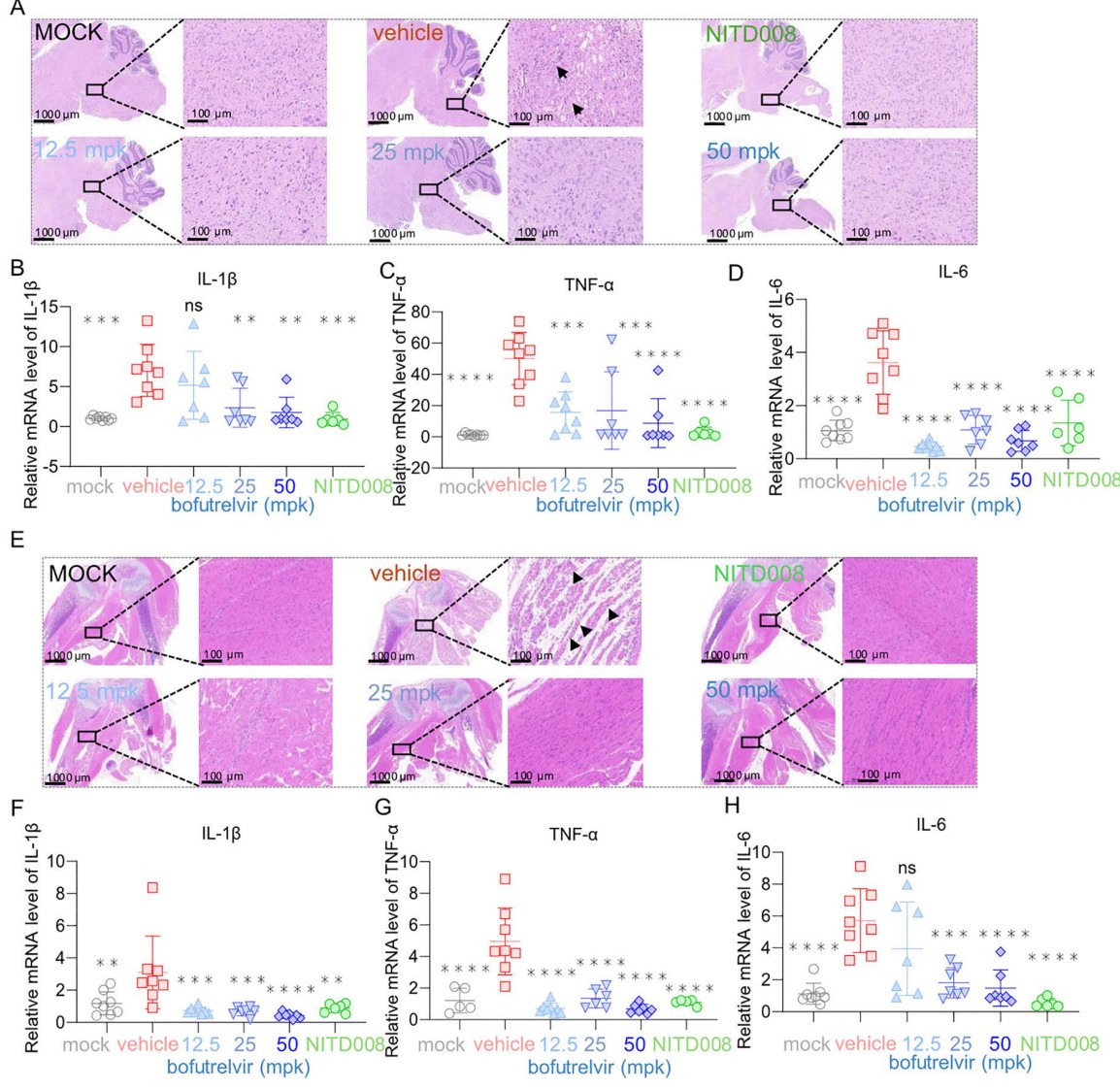

**Fig 5. Bofutrelvir attenuates EV71-induced neuropathology and inflammation in brain and muscle tissues. A.** Representative H&E-stained brain sections from EV71-infected neonatal mice treated with vehicle, bofutrelvir (12.5, 25, or 50 mpk, i.p.), or NITD008 (5 mpk, p.o.). Vehicle-treated mice exhibited typical histopathological signs of neuroinflammation, including neuronal degeneration and immune cell infiltration (arrowheads), which were alleviated by bofutrelvir. **B–D.** Expression levels of pro-inflammatory cytokines IL-1β, TNF-α, and IL-6 in brain tissue, as measured by RT-qPCR. Bofutrelvir treatment significantly reduced cytokine levels compared to the vehicle group. **E.** Representative H&E-stained hind limb muscle tissue sections from EV71-infected neonatal mice treated with vehicle, bofutrelvir (12.5, 25, or 50 mpk, i.p.), or NITD008 (5 mpk, p.o.). Histopathological analysis of hind limb muscle tissue. Vehicle-treated mice displayed severe muscle fiber damage and inflammatory infiltration. **F–H.** Expression levels of IL-1β, TNF-α, and IL-6 in muscle tissue, showing dose-dependent suppression of local inflammation following bofutrelvir administration. Data are shown as mean±SD from independent biological replicates (n=6-8). Scale bars, 1000 μm. Statistical significance was compared between the vehicle group and test groups, which were represented by asterisks marked correspondingly in the figure. (* $P<0.05$ and ** $P<0.01$, *** $P<0.001$ **** $P<0.0001$ by non-parametric One-way ANOVA analysis, respectively).

dose-dependent manner. Plaque assay results showed a corresponding decrease in infectious viral titers across the same tissues (Fig 6F-6H). Notably, oral bofutrelvir maintained strong antiviral activity in the spinal cord and hind limb muscles. Consistent with these findings, survival analysis revealed that vehicle-treated mice experienced substantial

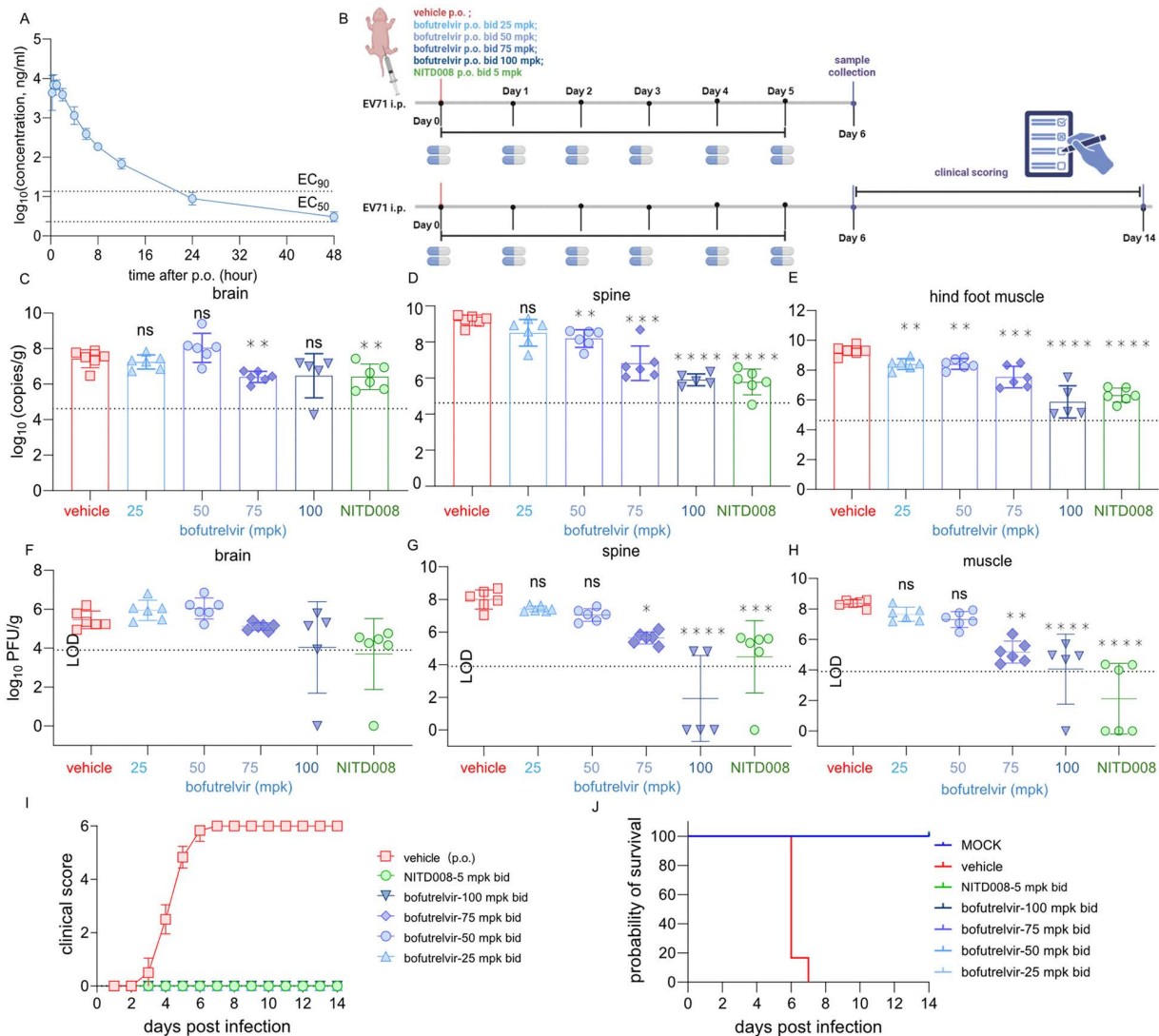

**Fig 6. Efficacy of orally administered bofutrelvir confirms its potential as a viable therapeutic against EV71. A.** Pharmacokinetic profile of bofutrelvir following a single oral dose (25 mpk) in beagle dogs. Plasma concentrations over time show favorable oral exposure and support dosing strategies for subsequent in vivo evaluation, with calculated parameters (including $C_{max}$ and $T_{1/2}$ presented as mean ± SD. Dashed horizontal lines indicate the in vitro $EC_{50}$ and $EC_{90}$ values of bofutrelvir against EV71, highlighting that plasma levels remain above the therapeutic threshold for an extended period. **B.** Experimental Flow Chart. A schematic diagram illustrating the design of the oral efficacy study in neonatal mice. Six-day-old ICR mice were orally infected with EV71 ($1 \times 10^6$ PFU) and subsequently treated by oral gavage with bofutrelvir at varying doses from day 0 to day 5. Mice were monitored daily for clinical symptoms, and samples were collected at the designated endpoints. Figure Created in BioRender. Cao, **J.** (2026) https://BioRender.com/yo37i97. C-E. EV71 Viral RNA Copy Numbers in Brain, Spinal Cord, and Hind Foot Muscle. RT-qPCR quantification demonstrated a marked reduction in viral RNA levels in the brain **(C)**, spinal cord **(D)**, and hind foot muscle **(E)** of bofutrelvir-treated mice compared to vehicle controls. F-H. Infectious Viral Titers in Tissues. Infectious EV71 titers were determined by plaque assay in the brain **(F)**, spinal cord **(G)**, and hind foot muscle **(H)**. Data are presented as mean ± SD (n = 6–8 per group). Statistically significant differences between the vehicle and treatment groups were evaluated by non-parametric one-way ANOVA and are denoted as follows: *$P < 0.05$; **$P < 0.01$; ***$P < 0.001$; ****$P < 0.0001$. I-J. Clinical scores and survival rates of neonatal ICR mice infected with EV71. Groups of 6 mice were infected via the i.p. route with $10^6$ pfu of EV71. Drug treatment started on the day of infection and consisted of oral administration b.i.d. of bofutrelvir, NITD008 or the vehicle alone for 5 consecutive days. The mice were monitored for 14 days. Disease severity was assessed using a standardized scoring system: 0, normal; 1, ruffled hair and hunchbacked appearance; 2, limb weakness; 3, paralysis in one limb; 4, paralysis in two limbs; 5, inability to move or feed (mice reaching this score were humanely euthanized, with subsequent scores recorded as described); and 6, death.

mortality between days 6 and 7 post infection, whereas bofutrelvir conferred robust protection against EV71-induced lethality (Fig 6J). All mice treated with ≥25 mpk bofutrelvir survived the infection, achieving 100% survival, similar to the mock and NITD008 control groups.

To further evaluate the breadth of oral efficacy, we established a parallel CA16 neonatal mouse model. Mice infected intraperitoneally with $10^6$ PFU of CA16 received oral bofutrelvir (25, 50, or 75 mpk, twice daily) from Day 0 to Day 5. Treatment resulted in dose-dependent clinical improvement and significant reductions in viral RNA across brain, spinal cord, and hind limb tissues (S4 Fig). Notably, survival analysis demonstrated a clear dose-dependent protective effect: oral administration of bofutrelvir at 75 mpk completely protected infected mice from CA16-induced lethality, while partial but significant survival benefits were observed at 50 mpk and 25 mpk compared with vehicle-treated controls.

These data demonstrate that oral bofutrelvir confers both clinical and survival benefits in neonatal mice infected with EV71 and CA16, further supporting its broad-spectrum therapeutic potential against enteroviruses. Collectively, the consistent clinical protection and survival benefits observed across multiple neonatal enterovirus models support the strong translational potential of bofutrelvir for further clinical development.

## Discussion

In this study, we leveraged the conserved structural and catalytic features shared between enterovirus 3Cpro and coronavirus Mpro to explore a rational drug repurposing strategy. Although these proteases originate from distinct viral families, they exhibit similar overall folds, a cysteine-centered catalytic mechanism, and strict substrate specificity for P1-glutamine residues, providing a mechanistic basis for cross-protease inhibition. Through target biochemical screening, we found that bofutrelvir potently inhibits the viral 3Cpro, an essential enzyme for polyprotein processing and viral replication. The compound demonstrated nanomolar antiviral activity across various cell lines, effectively suppressed viral replication in neonatal mouse models, and demonstrated robust oral bioavailability, thereby supporting its potential for clinical repurposing.

Mechanistically, bofutrelvir covalently binds the catalytic cysteine of EV71 3Cpro, and our co-crystal structure confirms that its aldehyde warhead forms a covalent bond with the active site. The γ-lactam moiety-like P1 residue mimics the substrate's glutamine, while the compound induces a conformational shift in the β-ribbon, stabilizing it in a closed state—thereby blocking access to the substrate-binding cleft [21]. Although this inhibitory mode resembles that of the first-generation 3Cpro inhibitor rupintrivir, bofutrelvir features a distinct scaffold and demonstrates superior in vivo efficacy, highlighting the importance of optimized molecular architecture beyond target engagement alone.

Consistent with the shared protease features between enterovirus and coronavirus, bofutrelvir displayed potent inhibition of both EV71 3Cpro and SARS-CoV-2 Mpro. In contrast, other approved SARS-CoV-2 Mpro inhibitors, such as nirmatrelvir and ensitrelvir, exhibit limited activity against enteroviral proteases. This difference may reflect structural elements unique to bofutrelvir, including its unique P3 moiety, which likely facilitates better accommodation in structurally diverse protease pockets.

We also observed that several caspase-family inhibitors, such as Ac-YVAD-CMK, inhibited EV71 3Cpro *in vitro*. While this may reflect non-specific reactivity toward cysteine proteases due to shared electrophilic warheads, a more plausible explanation involves substrate sequence similarity. Nonetheless, such compounds lacked antiviral activity in cells, reinforcing that target engagement alone is insufficient without adequate bioavailability and intracellular stability, both of which are met by bofutrelvir.

A key strength of this study is the demonstration of broad-spectrum antiviral activity. Bofutrelvir effectively inhibited multiple enterovirus species, including *enterovirus alphacoxsackie, enterovirus betacoxsackie,* and echovirus 11, extending its utility beyond EV71. In vivo, oral administration of bofutrelvir resulted in dose-dependent antiviral effects in EV71-infected neonatal mice, with marked reductions in viral loads in peripheral tissues and the spinal cord. Although suppression of viral replication in the brain was less pronounced, likely due to limited CNS penetration, overall disease severity was significantly reduced. Similar protective effects were observed in a neonatal mouse model of CA16 infection, demonstrating

that oral bofutrelvir confers efficacy across clinically relevant enteroviruses (S4 Fig). These findings show that bofutrelvir is effective against multiple enteroviruses when given orally, supporting its potential as a broad-spectrum treatment. The neonatal mouse model used here replicates key features of severe human EV71 and CA16 infections—including neuroinvasion and limb paralysis—making it a highly relevant platform for preclinical drug testing. With no approved antivirals for these infections and their significant impact on young children, our findings position bofutrelvir as a strong candidate for further clinical development.

Bofutrelvir has previously undergone Phase I and II clinical evaluations for the treatment of SARS-CoV-2 during which it demonstrated a favorable safety and tolerability profile. No major adverse effects or dose-limiting toxicities were reported in these trials, supporting its suitability for clinical repurposing. The compound's covalent, yet reversible, mechanism of action enables potent viral inhibition while minimizing off-target effects, particularly when compared to broader-spectrum cysteine-reactive agents. Although those trials were not completed, our current results demonstrate that oral formulation is feasible and effective in the context of enteroviral infections. This shift toward oral delivery markedly improves patient accessibility and expands its potential indications. In this context, pharmacokinetic properties critically differentiate bofutrelvir from earlier enterovirus 3C$^{pro}$ inhibitors such as rupintrivir. Despite potent in vitro activity, rupintrivir exhibits extremely poor systemic exposure following oral or intranasal administration, which limited its clinical applicability to localized infections [51–53]. These findings underscore that improved drug-like properties, beyond target engagement alone, are essential for the successful translation of enterovirus protease inhibitors into clinically viable therapies.

Concerns regarding aldehyde-containing warheads have historically limited their consideration in drug development; however, clinical experience indicates that aldehyde moieties can be well tolerated when incorporated into stable molecular scaffolds. Several approved antibiotics and clinical-stage compounds support the feasibility of this approach. Consistent with this principle, bofutrelvir employs a reversible aldehyde warhead and has demonstrated good safety and tolerability in human studies, further supporting the translational potential of aldehyde-based antiviral agents.

Serial passaging of EV71 in the presence of sub-inhibitory concentrations of bofutrelvir for up to 16 generations did not result in the emergence of resistant viral phenotypes (data as shown in S5 Fig). While compared to NITD008, some limitations remain. The limited CNS penetration of oral bofutrelvir reduces its efficacy against EV71-associated neurological disease. Future efforts might aim to enhance brain delivery through prodrug strategies or formulation technologies.

In conclusion, this study establishes bofutrelvir as a potent, orally available, structurally validated 3C$^{pro}$ inhibitor with broad-spectrum efficacy against EV71 and other enteroviruses. Its favorable pharmacokinetics, robust antiviral activity, and structural rationale support its further development as a next-generation antiviral candidate for enteroviruses infections.

## Materials and methods

### Ethics statement

The ICR neonatal mice were housed under specific pathogen-free (SPF) conditions in the facilities provided and supported by Animal Resource Center of Wuhan Institute of Virology, Chinese Academy of Sciences. All animal experiments were performed in Animal Biosafety Level 2 Laboratory (ABSL-2). All animal experiments were carried out in strict accordance with the National Institute of Health guidelines under protocols approved by the Institutional Animal Care and Use Committee of Wuhan Institute of Virology (the approval number: No. WIVA25202307). The study adhered to national and institutional guidelines for animal care, and protocols were approved by the Institutional Animal Care and Use Committee (IACUC).

### Cell and virus

The African green monkey kidney (Vero) cells, human rhabdomyosarcoma (RD) cells and Cancer coli-2(Caco-2) cells were grown in Dulbecco's modified Eagle's medium (DMEM, GIBCO) supplemented with 10% fetal bovine serum (FBS, GIBCO) and 1% penicillin–streptomycin antibiotics. U251 cells was maintained in Minimum Essential Medium (MEM) with

10% fetal bovine serum (FBS) and 1% penicillin–streptomycin antibiotics. All cells were cultured with 5% $CO_2$ at 37°C. EV71 (strain BrCr, CSTR: 16533.06. IVCAS 6.6305) was obtained from the Microorganisms & Viruses Culture Collection Center, Wuhan Institute of Virology (WIV), Chinese Academy of Sciences (CAS) and amplified in RD cells. EV71 (strain GZ-CII) was kindly supplied by Prof. Yi Xu from Guangzhou Medical University, Guangzhou [50]. CA16(strain GZ/0301/2011), CA6(ATCC: VR-181), CA10(ATCC: VR-168), CB5(ATCC: VR-185) were provided by Hecin Scientific Company (Guangzhou, China) and amplified in RD cells. All the viruses were stored as aliquots at ≤ -80°C, and used for antiviral activity assays.

## Chemical reagents

A focused library of 104 peptidomimetic protease inhibitors was purchased from MedChemExpress (MCE, Monmouth Junction, NJ, USA). The library compounds were dissolved in DMSO to a stock concentration of 40 mM and stored at –80 °C until use.

The compound bofutrelvir and its corresponding formulation were kindly provided by Frontier Biotech Co., Ltd. (Nanjing, China).

## Protein expression and purification

His-tagged EV71-3C$^{pro}$ with a Tobacco Etch Virus (TEV) Protease cleavage site was cloned into a pET28-b vector. The resulting plasmids were transformed into BL21 (DE3) cells for protein expression. Overnight cultures in kanamycin media were grown from single colonies. Each culture was used to inoculate 1 L cultures in LB, supplemented with 50 μg/mL kanamycin. These cultures grew in Fernbach flasks at 37 °C while shaking at 180 rpm, until the OD600 reached approximately 0.75, at which point the temperature was reduced to 16 °C, 0.3 mM IPTG was added, and they were left to grow overnight.

For protein purification, cells were pelleted and then resuspended in lysis buffer (20 mM Tris-HCl [pH 8.0], 150 mM NaCl) and disrupted by a JN-02C high-pressure homogenizer (Guangzhou Juneng Nano & Bio Technology Co., Ltd). The expressed proteins were purified by a Ni-NTA column (GE) and transformed into the cleavage buffer (20 mM Tris, pH 8.0, 150 mM NaCl, 2 mM DTT) containing TEV Protease for removing the 6 × His tag. The resulting proteins were further purified by size exclusion chromatography using a Superdex 200 column (GE Healthcare). The purified proteins were concentrated to 15–25 mg/mL, flash frozen in liquid nitrogen, and stored as aliquots at -80°C.

## Enzymatic activity and inhibition assays

A fluorescence resonance energy transfer (FRET) protease assay was applied to measure the inhibitory activity of compounds against the EV71 3C$^{pro}$. The fluorogenic substrate NMA-IEALFQGPPK(DNP)FR was synthesized by Motif biotech (Suzhou, China) with a purity of 90%. The assay was carried out in 384-well plates with 20 μL of reaction mixtures containing the reaction buffer (50 mM Tris-HCl pH 7.0, 150mM NaCl,1mMEDTA, 1mMDTT, 10% glycerol), 1 μM EV71 3C$^{pro}$ and the tested compounds dissolved in DMSO. The fluorogenic substrates were added to initiate the reaction. Hydrolysis of the substrate was monitored with SYNERGY H1 (Bio Tek, USA) at excitation and emission wavelengths of 340 and 500 nm at 30 °C for 1 hour. Inhibition of EV71 3C$^{pro}$ activity of the tested compounds was calculated using the formula $(V_{DMSO}-V_{tested\ compounds})/V_{DMSO}*100\%$. Values are averaged from three independent experiments. For IC$_{50}$ determination of the tested compounds, the same reaction was performed as described above with serial dilutions of the inhibitors as indicated, and IC$_{50}$ value was calculated by using Graphpad Prism 9.

## High-throughput inhibitor screen

As previously described, high-throughput screening was conducted using the FRET assay on EV71 3C$^{pro}$. The first round of screening for each compound was conducted with a final concentration of 100 μM. For those with an inhibition rate of EV71 3C$^{pro}$ higher than 90%, double gradient dilution will be conducted for re-screening to obtain IC$_{50}$.

## Compound effect inhibition and cytotoxicity assays

To evaluate the antiviral activity of compounds, RD cells were seeded in 48-well plates (30,000 cells/well) and incubated overnight. The next day, culture medium was replaced with compound-containing medium at gradient concentrations for 45 minutes. Cells were then infected with EV71 at a multiplicity of infection (MOI) of 0.005. After 1 h of viral adsorption, the supernatant was removed and replaced with fresh compound-containing medium, followed by incubation for 24 h.

To assess the antiviral effect of bofutrelvir across different cell types, Vero, Caco-2, and U251 cells were infected with EV71 at MOIs of 0.01, 1, and 0.005, respectively. After treatment, viral RNA was extracted from the supernatant using a commercial viral RNA extraction kit (Vazyme, China). RT-qPCR was performed using a HiScript II One Step RT-qPCR SYBR Green Kit (Vazyme, China) with EV71-specific primers (forward: 5'-ATTATCCGACCCACCAGCAC-3'; reverse: 5'-CGAGGTATCCACGCTCTGAC-3'). Viral inhibition was calculated based on RNA copy numbers, and the $EC_{50}$ values were determined using GraphPad Prism 9. For CA16, CA6, CA10, and CVB5, a universal enterovirus primer pair (forward: 5'-TCCTCCGGCCCCTGA-3'; reverse: 5'-AATTGTCACCATAAGCAGCCA-3') was used. Standard curves were generated using serial dilutions of plasmids encoding the EV71 VP1 fragment (nt 550–800, cloned into pCE2; Vazyme) or a conserved enterovirus 5′UTR region (nt 300–600, cloned into pUC57; Sangon Biotech).

Viral titers were assessed using a $TCID_{50}$ assay. Vero cells were seeded in 96-well plates (10,000 cells/well) and incubated for 16 h. Supernatants from compound-treated plates were serially diluted ($10^1$–$10^6$-fold) in DMEM with 2% FBS and added to cells in six replicates per dilution. Plates were incubated at 37 °C with 5% $CO_2$ for 5 days. Cytopathic effects (CPE) were recorded, and $TCID_{50}$ values were calculated using the Reed–Muench method.

For immunofluorescence assays, infected cells were fixed with 4% paraformaldehyde, washed with PBS, and blocked with 5% skim milk containing 0.2% Triton X-100 for 1 h. Cells were then incubated overnight at 4 °C with mouse anti-EV71 VP1 monoclonal antibody (Abcam, clone 10F0, 1:1000), followed by incubation with CoraLite 488-conjugated goat anti-mouse IgG (Proteintech, China). Nuclei were stained with DAPI (Beyotime, China), and images were captured using a CYTATION H1 imaging system (BioTek, USA).

Cell viability was measured using a Cell Counting Kit-8 (CCK-8; Beyotime, China). RD, Vero, Caco-2, and U251 cells were seeded in 96-well plates (10,000 cells/well), treated with compounds (200 to 6.25 μM) for 24 h, and analyzed according to the manufacturer's protocol. Absorbance was read at 450 nm using a SYNERGY H1 plate reader (BioTek, USA), and viability was calculated relative to vehicle-treated controls. All experiments were independently repeated at least twice, each in triplicate. Data are presented as mean ± SD.

## Protein crystallization and structure determination

Crystals used for seeding were grown by thawing protein on ice and diluting it to 2 mg/mL prior to the addition of bofutrelvir dissolved in DMSO. The final molar ratio of protease to inhibitor was 1:3. After 2 hours incubation on ice, the mixtures were clarified by centrifugation prior to the crystallization trails. Crystallization was performed at 16 °C using a sitting drop vapor diffusion method, by mixing equal volumes (0.3:0.3 μL) of the protein and crystallization solution. Crystals were finally yielded in a solution containing 0.2 M Trimethylamine N-oxide dihydrate, 0.1 M Tris (pH 8.5), 20% Polyethylene glycol monomethyl ether 2,000. Cryocooling was achieved by soaking the crystals for 30–60 s in reservoir solution containing 20% glycerol and by flash-freezing them in liquid nitrogen. X-ray diffraction data were collected at beamline BL19U1 at the Shanghai Synchrotron Radiation Facility (SSRF) at 100 K and at a wavelength of 0.9785 Å using a Pilatus3 6M image plate detector.

For structure determination, the diffraction data sets were indexed, integrated, and scaled by HKL2000 software packages. The complex structure was solved by molecular replacement (MR) using the program PHASER with a search model of PDB code 7DNC. The structure refinement was performed iteratively with manual model building in Coot and automated refinement by PHENIX. The 3,500-K composite simulated-annealing omit 2Fo-Fc electron density maps were generated by Crystallography & NMR System.

## EV71 Animal experiment

The pregnant mice with the same expected delivery date were acclimated in individually ventilated cages in an SPF environment under standard conditions. On the fifth day after birth, neonatal mice were divided randomly into different groups. For challenge studies, ICR neonatal mice were inoculated intraperitoneally with $1 \times 10^6$ PFU of EV71. Two hours after viral infection, mice were orally treated with vehicle, or drugs as described above. From then on, recorded daily body weight of the mice as well as clinical score every day. Clinical illness was scored as follows: 0, normalcy; 1, ruffled hair and hunchbacked appearance; 2, limb weakness; 3, paralysis in one limb; 4, paralysis in two limbs; 5, lose the ability to move and ingest; 6, death. As mice with a clinical score of 5 usually die in 1 day. For RT-qPCR assay tissue from excised organs On Day 6, anesthetized mice by isoflurane inhalation and then collecting brains, spines and hind foot muscles, weighed, and stored at ≤ -80°C.

## RT-qPCR assay and titration of virus from excised organs

The tissues were homogenized in pre-filled bead tubes using an automated homogenizer (Fisherbrand, 15-340-164), and followed by centrifugation (5,000 g for 10 minutes at 4°C). The clarified tissue homogenate supernatants were extracted RNA by a viral DNA/RNA extraction kit (Vazyme, China). RT-qPCR assay (universal primers for enterovirus sequences: forward primer 5'-TCCTCCGGCCCCTGA-3' and reverse primer 5'-AATTGTCACCATAA GCAGCCA-3') was performed using a HiScript II One Step RT-qPCR SYBR Green Kit (Vazyme, China) following the manufacturer's protocol. The relative amount of cytokine mRNA was calculated by the comparative threshold cycle method with GAPDH as control. The following primer pairs were designed according to sequences reported in GeneBank and used as follows: IFN-γ (forward primer: 5'-AAAGAGATAATCTGGCTCTGC-3' and reverse primer 5'-GCTCTGAGACAATGAACGCT-3'); TNF-α (forward primer: 5'- CCACCACGCTCTTCTGTCTAC-3' and reverse primer 5'-AGGGTCTGGGCCATAGAACT-3'); IL-1β (forward primer: 5'-TTGACGGACCCCAAAAGATG-3' and reverse primer 5'- AGAAGGTGCTCATGTCCTCA-3'); IL-6 (forward primer: 5'- GTTCTCTGGGAAATCGTGGA-3' and reverse primer 5'- TGTACTCCAGGTAGCTATGG-3'); GAPDH (forward primer: 5'- TGGTGAAGGTCGGTGTGAAC-3' and reverse primer 5'- GAAGGGGTCGTTGATGGCAA-3').

## Histopathology and immunohistochemistry

In brief, the tissues of infected mice were fixed in 4% paraformaldehyde for more than 7 days. The fixed samples were then embedded in paraffin wax and sectioned by a desktop tissue processor. The slices were dried and fixed on slides overnight at 37 °C. Then, for dewaxing, the slices were successively washed in environmentally friendly dewaxing solution and anhydrous ethanol three times. Fixed tissue samples were used for hematoxylin–eosin (H&E). Immunofluorescence staining for the detection of the EV71 VP1 (Anti-Enterovirus 71 antibody, Abcam) and nucleus (DAPI). The image information was collected using a Pannoramic MIDI system (3DHISTECH, Budapest) and FV1200 confocal microscopy (Olympus).

## Beagle dog pharmacokinetic study design

The pharmacokinetics of bofutrelvir was evaluated in healthy female beagle dogs. Dogs (body weight almost 10 kg) were fasted overnight for approximately 12 hours before dosing, with free access to water. bofutrelvir was administered as a single oral dose of 25 mpk via gavage. The formulation consisted of an aqueous suspension containing 5% DMSO + 5% anhydrous ethanol + 40% PEG 400 + 50% saline to ensure uniform dispersion. Each dog received the dose using a sterile gastric feeding tube under gentle restraint. Blood samples (1.5 mL) were collected from the cephalic vein at pre-dose (0 h) and at 0.25, 0.5, 1, 2, 4, 6, 8, 12, 24, 48 and 72 hours post-dosing. Blood was transferred into tubes containing EDTA-K2 anticoagulant and centrifuged at 3,000 × g for 10 minutes at 4°C to separate plasma. Plasma samples were aliquoted and stored at −80°C until analysis.

Plasma concentrations of bofutrelvir were measured using a validated LC-MS/MS method. Pharmacokinetic parameters, including $C_{max}$, $T_{max}$, $AUC_{0-t}$, $AUC_{0-\infty}$, $T_{1/2}$, and oral bioavailability, were calculated using non-compartmental analysis with WinNonlin (Certara, Princeton, NJ, USA). The pharmacokinetic data supported dose selection for subsequent in vivo efficacy studies.

## Statistics

All the statistical analysis and corresponding graphs were generated with GraphPad Prism 9. The data are presented as the mean± SD. One-way ANOVA (analysis of variance) with Dunnett's post-hoc test was used to determine statistical significance (*$P < 0.05$; **$P < 0.01$; ***$P < 0.001$; ****$P < 0.0001$; ns no significance.)

## Supporting information

**S1 Table. High-throughput screening identifies potent inhibitors Data.**
(XLSX)

**S1 Fig. Time-of-addition analysis of bofutrelvir against EV71 replication.** A. Schematic representation of the time-of-addition protocols used to dissect the mode of antiviral action. Four treatment strategies were employed. PRE (i): Cells were pre-treated with bofutrelvir for 2 h prior to viral infection. SIM (ii): Bofutrelvir and EV71 were simultaneously added to cells at time 0 h. DIR (iii): Virus was pre-incubated with bofutrelvir for 1 h, then diluted 500-fold before infecting the cells. POST (iv): Bofutrelvir was added 1 h after viral infection.B. Quantification of antiviral activity in each condition was assessed by RT-qPCR at 16 h post-infection. Strong inhibition was observed only in the POST condition, indicating that bofutrelvir primarily acts at a post-entry step, likely targeting viral replication. Data are presented as mean±SD from biological replicates (n=4).
(DOCX)

**S2 Fig. Structure Comparison of the inhibitor binding mode in EV71 3Cpro A.** Comparison of EV71 3Cpro bound to bofutrelvir (blue) and AG7088 (pink)(PDB: 3QZR), showing highly similar overall binding modes within the substrate-binding cleft. B. Close-up of the S2 sub-pocket. The cyclohexyl P2 group of bofutrelvir (blue) inserts deeper into the S2 pocket, whereas the aromatic P2 group of AG7088 (pink) is positioned closer to the pocket entrance. C. Comparison at the S4 region. AG7088 (pink) extends further toward the surface groove via its P3/P4 substituent, while bofutrelvir (blue) adopts a more compact conformation in this region.
(DOCX)

**S3 Fig. Dose-dependent inhibition of EV71 VP1 protein expression by bofutrelvir in mouse brain and hind limb muscle tissues.** A. Immunofluorescence staining of EV71 VP1 protein (green) and nuclei (blue, DAPI) in brain tissues. Bofutrelvir treatment resulted in dose-dependent reduction of VP1 expression. B. Immunofluorescence analysis of hind limb muscle tissue showing decreased VP1 levels in mice treated with bofutrelvir or NITD008 compared to vehicle controls.
(DOCX)

**S4 Fig. Bofutrelvir orally administered suppresses CA16 replication and pathogenesis in a neonatal mouse model.** A. Schematic of the in vivo experimental design.3 days old neonatal ICR mice were infected with CA16 via intra-peritoneal (i.p.) injection $10^6$ pfu on Day 0 (n=4–6) and treated twice daily with bofutrelvir (25,50 or 75 mg/kg, p.o.) or the positive control compound NITD008 (5 mg/kg, p.o.) from Day 0 to Day 5. Samples were collected on Day 6. Figure Created in BioRender. Cao, J. (2026) https://BioRender.com/yo37i97. B–D. Quantification of CA16 viral RNA copies numbers in brain, spinal cord, hind foot muscle, and hind foot tissues by qRT-PCR. (* $P < 0.05$ and ** $P < 0.01$, *** $P < 0.001$ **** $P < 0.0001$ by non-parametric One-way ANOVA analysis, respectively)E-F. Clinical scores and survival rates of neonatal

ICR mice infected with CA16. Groups of 5 mice were infected via the i.p. route with $10^6$ pfu of CA16. Drug treatment started on the day of infection and consisted of oral administration b.i.d. of bofutrelvir, NITD008 or the vehicle alone for 6 consecutive days. The mice were monitored for 14 days. (Clinical illness was scored as follows: 0, normalcy; 1, ruffled hair and hunchbacked appearance; 2, limb weakness; 3, paralysis in one limb; 4, paralysis in two limbs; 5, lose the ability to move and ingest; 6, death. As mice with a clinical score of 5 usually die in 1 day, they were euthanized with carbon dioxide to reduce their suffering, and their death days were accounted on the next day.).
(DOCX)

**S5 Fig. Selection and characterization of bofutrelvir-resistant EV71.** A. Scheme used for selection of bofutrelvir-resistant EV71. The concentrations of bofutrelvir applied at different passage numbers during the selection process indicated. B. Characterization of resistance phenotypes of P16 EV71. The indicated concentrations of bofutrelvir were added to RD cells after 1h of infection with EV71 at MOI=0.005. The viral titers in culture fluids were determined by qRT-PCR assay.
(DOCX)

**S1 Raw Data. All data generated and analyzed in this study.**
(XLSX)

## Acknowledgments

We thank Xuefang An in the Center for Experimental Animals and Ding Gao in the Center for Instrumental Analysis and Metrology of the Wuhan Institute of Virology for their technical assistance. We thank Wenhua Kuang and Zhenhua Tian for protein crystallization and structure determination.

## Author contributions

**Conceptualization:** Hong Liu, Junyuan Cao, Lei-Ke Zhang.

**Data curation:** Zhengyu Ye, Shaolin Zhang.

**Formal analysis:** Zhengyu Ye, Shaolin Zhang, Junyuan Cao.

**Funding acquisition:** Gengfu Xiao, Hong Liu, Lei-Ke Zhang.

**Investigation:** Zhengyu Ye, Wenhao Dai, Shaolin Zhang, Yingchun Xiang, Jinlin Wang, Yumin Zhang, Wenyuan Cao, Zuyi Li, Fan Feng.

**Methodology:** Zhengyu Ye, Shaolin Zhang, Junyuan Cao.

**Project administration:** Junyuan Cao, Lei-Ke Zhang.

**Resources:** Junyuan Cao, Lei-Ke Zhang.

**Supervision:** Junyuan Cao, Lei-Ke Zhang.

**Validation:** Zhengyu Ye, Junyuan Cao, Lei-Ke Zhang.

**Visualization:** Zhengyu Ye, Shaolin Zhang, Junyuan Cao.

**Writing – original draft:** Zhengyu Ye, Shaolin Zhang, Junyuan Cao.

**Writing – review & editing:** Johan Neyts, Gengfu Xiao, Hong Liu, Lei-Ke Zhang.

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
