## [Decision Letter · Decision Letter 0]

2 Dec 2025

PPATHOGENS-D-25-02218

Repurposing of a Clinical Protease Inhibitor Enables Oral, In Vivo Broad-Spectrum Inhibition of Enteroviruses

PLOS Pathogens

Dear Dr. Zhang,

Thank you for submitting your manuscript to PLOS Pathogens. After careful consideration, we feel that it has merit but does not fully meet PLOS Pathogens's publication criteria as it currently stands. Therefore, we invite you to submit a revised version of the manuscript that addresses the points raised during the review process.

We look forward to receiving your revised manuscript.

Kind regards,

George A. Belov, PhD

Academic Editor

PLOS Pathogens

Alexander Gorbalenya

Section Editor

PLOS Pathogens

Sumita Bhaduri-McIntosh

Editor-in-Chief

PLOS Pathogens

orcid.org/0000-0003-2946-9497

Michael Malim

Editor-in-Chief

PLOS Pathogens

orcid.org/0000-0002-7699-2064

**Additional Editor Comments :**

- Please refer to picornavirus (virus) taxa, including species according to ICTV rules on nomenclature

- Acknowledge key similarities between enterovirus 3Cpro and coronavirus Mpro (which also is known as 3C-like protease, 3CLpro) that are critical for the background of this study and discussion of its results. The manuscript presentation, including the Title, Abstract, Intro and Discussion and Reference list, must be improved to reveal gain of knowledge compared to the updated background. Below are two reviews to relevant literature

https://doi.org/10.1016/j.antiviral.2010.12.007
https://doi.org/10.1007/BF02174046

**Journal Requirements:**

**Reviewers' Comments:**

Reviewer's Responses to Questions

**Part I - Summary**

Reviewer #1: This manuscript presents a well-executed and highly relevant study that addresses the urgent need for antiviral therapies against enteroviruses. The authors convincingly demonstrate that bofutrelvir, originally developed as a SARS-CoV-2 Mpro inhibitor, potently inhibits enteroviral 3C protease activity and exhibits broad-spectrum antiviral efficacy against multiple enteroviruses, including EV71, Coxsackievirus A16, and Echovirus 11.

The combination of enzymatic, cellular, and in vivo studies, supported by structural elucidation of the inhibitor–protease complex, provides a comprehensive understanding of the mechanism of action and underscores the therapeutic potential of bofutrelvir. The demonstrated reduction of viral loads and alleviation of disease symptoms in infected neonatal mice further enhance the translational significance of the work.

Reviewer #2: In this manuscript, the authors screened a library of 104 protease inhibitors to search for new inhibitors of EV71 3C protease. They found bofutrelvir, a clinical stage peptidomimetic developed for SARS-CoV2 Mpro, has potent antiviral activity against multiple enteroviruses in cell culture. Crystal structure of bofutrelvir binding to EV71 3C protein was resolved, supporting a mechanism of covalently inhibiting the protease active site. Bofutrelvir demonstrated in vivo efficacy against EV71 infection in neonatal mice through intraperitoneal injection or oral administration, and against CA16 infection through oral administration. The data are convincing, and the experiments are well performed. The discovery of bofutrelvir as a broad-spectrum enterovirus 3C inhibitor provides the foundation for repurposing this compound for antivirals against enterovirus diseases.

**Part II – Major Issues: Key Experiments Required for Acceptance**

Reviewer #1: bofutrelvir was previously reported by authors as enterovirus 3CLpro inhibitors. This needs to be updated and clearly spelled out in the introduction.

Reviewer #2: What is the survival curve of the neonatal mice after EV71 and CA16 infection? does bofutrelvir provide any protection against mortality?

The author showed that the efficacy of bofutrelvir in cell culture is comparable to or even better than that of rupintrivir, the known enterovirus 3C inhibitor. Comparison beyond this point is lacking. Rupintrivir has failed clinical trials. It is not clear from the manuscript whether bofutrelvir has any advantage over rupintrivir. The author can use their own experimental data or extrapolate from published dataset for such comparisons, which may include:

1. How does the bofutrelvir-3C structure compare to rupintrivir-3C structure? The author briefly mentions that the two structures are similar. A more detailed comparison should be included.

2. what is the in vivo efficacy and dose range of rupintrivir in the neonatal mouse model, or other in vivo model?

3. how does the bioavailability of bofutrelvir and rupintrivir compare?

**Part III – Minor Issues: Editorial and Data Presentation Modifications**

Reviewer #1: "Several antiviral agents targeting different stages of the EVs life cycle have been investigated"

Comment: reviews of EV-A71 and EV-D68 antivirals should be updated

Acta Pharm. Sin. B. 2022, 12, 1542-1566.

Lab Invest. 2024 Feb;104(2):100298.

ACS Infect. Dis. 2020, 6, 1572-1586.

Viruses 2025, 17(9), 1178

"One possible explanation for this discrepancy is that these compounds preferentially bind to

139 endogenous caspase family proteases within the host cells, which share similar catalytic cysteine

140 residues"

Comment: another possibility might be the lack of cellular permeability due to the negative charge of the aspartic acid.

Discussion session should be updated with the translational potential of aldehyde containing drugs.

Reviewer #2: Fig.2B what is the estimated EC50 according to the TCID50 assay? How does it compare to the qPCR assay in Fig.2A?

Why were clinical scores in Fig. 4 a lot less than Fig.6?

Line 282: “with strong antiviral effects seen in peripheral tissues such as the spinal cord and hind limb muscles”. Spinal cord is not a peripheral tissue.

PLOS authors have the option to publish the peer review history of their article (what does this mean? ). If published, this will include your full peer review and any attached files.

**Do you want your identity to be public for this peer review?** For information about this choice, including consent withdrawal, please see our Privacy Policy .

Reviewer #1: No

Reviewer #2: No

**Figure resubmission:**
---

## [Editor Report · Decision Letter 1]

5 Feb 2026

PPATHOGENS-D-25-02218R1

Repurposing of a Clinical Protease Inhibitor Enables Oral, In Vivo Broad-Spectrum Inhibition of Enteroviruses

PLOS Pathogens

Dear Dr. Zhang,

Thank you for submitting your manuscript to PLOS Pathogens. After careful consideration, we feel that it has merit but does not fully meet PLOS Pathogens's publication criteria as it currently stands. Therefore, we invite you to submit a revised version of the manuscript that addresses the points raised during the review process (see below signature line).

We look forward to receiving your revised manuscript.

Kind regards,

George A. Belov, PhD

Academic Editor

PLOS Pathogens

Alexander Gorbalenya

Section Editor

PLOS Pathogens

Sumita Bhaduri-McIntosh

Editor-in-Chief

PLOS Pathogens

orcid.org/0000-0003-2946-9497

Michael Malim

Editor-in-Chief

PLOS Pathogens

orcid.org/0000-0002-7699-2064

**Additional Editor Comments:**

Please fully address the editorial suggestions from the previous review. Specifically:

1)  Improve accuracy and detail of the title, e.g. " A clinical orally-available SARS-CoV-2 Mpro inhibitor blocks replication of multiple enteroviruses in cell culture and controls enterovirus A71 infection in animal models" ;

2) Restructure Abstract and place the known link between 3Cpro and Mpro BEFORE your screening results;

3) Author Summary. Replace COVID-19 with SARS-CoV-2. Specify the known link between picornavirus 3Cpro and coronavirus Mpro that made success of your research feasible;

4) Revisit the logic and gain of knowledge of your study in the Introduction and Discussion;

5) Please check with ICTV web-site if names of species *enterovirus A-D*  and *rhinovirus A-C*  remain actual and, if necessary, update the text accordingly;

6) Please reverse edits of the first sentence of the Introduction and keep plural form for enteroviruses and rhinoviruses (there are many of those);

7) Consider acknowledging the *Picornaviridae*  Study Group for the *Picornaviridae*  taxonomy and foundational research on enterovirus/picornavirus molecular biology;

8) Highlight changes to the Ref list;

**Reviewers' Comments:**

**Figure resubmission:**
---

## [Editor Report · Decision Letter 2]

2 Mar 2026

Dear Dr. Zhang,

We are pleased to inform you that your manuscript 'A Clinical SARS-CoV-2 Mpro Inhibitor Blocks Replication of Multiple Enteroviruses and Confers Oral In Vivo Protection in Animal Models' has been provisionally accepted for publication in PLOS Pathogens.

Best regards,

George A. Belov, PhD

Academic Editor

PLOS Pathogens

Alexander Gorbalenya

Section Editor

PLOS Pathogens

Sumita Bhaduri-McIntosh

Editor-in-Chief

PLOS Pathogens

orcid.org/0000-0003-2946-9497

Michael Malim

Editor-in-Chief

PLOS Pathogens

orcid.org/0000-0002-7699-2064
---

## [Editor Report · Acceptance letter]

Dear Dr. Zhang,

We are delighted to inform you that your manuscript, "A Clinical SARS-CoV-2 Mpro Inhibitor Blocks Replication of Multiple Enteroviruses and Confers Oral In Vivo Protection in Animal Models," has been formally accepted for publication in PLOS Pathogens.

Best regards,

Sumita Bhaduri-McIntosh

Editor-in-Chief

PLOS Pathogens

orcid.org/0000-0003-2946-9497

Michael Malim

Editor-in-Chief

PLOS Pathogens

orcid.org/0000-0002-7699-2064